# INVARIANCE AS A NECESSARY CONDITION FOR ONLINE CONTINUAL LEARNING

## ABSTRACT

Traditional supervised learning learns only enough features that are sufficient to classify the current given classes. It may not capture all the characteristics of the data. This is problematic for online continual learning (OCL), which learns a sequence of tasks incrementally, as it is a major cause for *catastrophic forgetting* (CF). Although numerous OCL methods have been proposed to mitigate CF, theoretical understanding of the problem is limited. Recent work showed that if the OCL learner can learn as many features as possible from the data (dubbed *holistic representations*), CF can be significantly reduced. This paper shows that learning *holistic representations* is insufficient and it is also necessary to learn **invariant representations** because many features in the data are irrelevant or variant, and learning them may also cause CF. This paper studies it both theoretically and empirically. A novel invariant feature learning method related to *causal inference theory* is proposed for online CL, which boosts online CL performance markedly.[1]

## 1 INTRODUCTION

A major challenge of continual learning (CL) is *catastrophic forgetting* (CF) (McCloskey & Cohen, 1989), which is caused by updating the parameters learned from previous tasks when learning a new task. This paper focuses on overcoming the CF problem in the challenging *online class-incremental learning* (online CIL) setting of CL. In CIL, a classifier is built for the classes learned thus far and no task-related information (task-id) is provided in testing.[2] In online CIL, the data come in a stream and the model visits each batch of training data only once, which makes learning new knowledge and protecting previous knowledge much harder. To mitigate CF in online CIL, many empirical replay-based techniques (Aljundi et al., 2019a; Prabhu et al., 2020; Guo et al., 2023) have been proposed and they mainly focus on the buffer storage management and rehearsal strategies. However, the **theoretical understanding** of the problem is still very limited.

This paper is concerned with **necessary conditions** for feature learning in the replay-based online CIL system. This paper proposes one necessary condition: learning and using features that are **invariant** for each class to form the class representation rather than features that are present in the input but are irrelevant to the class. This is **critical for online CIL** as variant features can cause serious CF in learning the classes of subsequent tasks. We explain it in the following example: In learning a task to classify images of *apple* and *fish*, some green background features are learned for apple, but these features are not invariant to apple. When a new class *cow* in a new task needs to be learned, these green background features are shared and may cause high logit outputs for apple and cow, which confuse the learner. The learner then has to modify the representations to reduce the logit value for apple, which causes CF. If the learner has learned the shape and other invariant features of the apple, the input of cow will not activate many parameters for apple. Then, in learning cow, changes to the parameters that are important for the apple will be limited, resulting in less CF.

This paper makes the following main contributions:

**(1).** It raises the *theoretical* and *critical* issue of **learning invariant features** for online CIL. It is critical for online CIL and also CIL because we learn new tasks incrementally and each new

---

[1]The code is included in the submitted supplemental material.

[2] The other popular CL setting is *task-incremental learning*, which provides the task-id for each test case.

task represents a major distribution shift and thus non-i.i.d. As our example above demonstrates, non-invariant features can cause serious CF for CIL and poor overall classification performances.

**(2).** It *theoretically analyzes* the invariance issue and **proves** that variant features can lead to higher CF (Sec. 4).

**(3).** It proposes a new online CIL method called **IFO** (*I*nvariant *F*eature learning for *O*nline CIL) based on *experience replay* to learn invariant features. IFO includes a new **invariant feature learning optimization objective** based on the guidance of the theoretical analysis and two approaches with three ways to create **environmental variants** to enable invariant feature learning. We justify IFO theoretically with **causal inference theory** (Sec. 5.4). Note that our environmental variants are different from existing data augmentations, which aim to diversify the full images, while IFO diversifies only the environments/backgrounds of the images (see Sec. 5.2 and Sec. 6.4).

**(4).** It empirically evaluates the proposed IFO in three online CIL scenarios: *traditional disjoint task*, *blurry task boundary*, and *data shift*. IFO improves the online CIL performance of the replay-based methods significantly by learning invariant feature representations. Combining IFO with another holistic representation learning method OCM boosts the performance further by a large margin.

## 2 RELATED WORK

Existing empirical approaches to CL can be grouped into several families. The *replay* approach saves a small amount of past data and uses it to adjust the previous knowledge in learning a new task Rebuffi et al. (2017); Wu et al. (2019); Hou et al. (2019); Chaudhry et al. (2020); Zhao et al. (2021); Korycki & Krawczyk (2021); Sokar et al. (2021); Yan et al. (2021); Wang et al. (2022a). *Pseudo-replay* generates replay samples Hu et al. (2019); Sokar et al. (2021). Using *regularizations* to penalize changes to important parameters of previous tasks is another approach Kirkpatrick et al. (2017); Ritter et al. (2018); Ahn et al. (2019); Yu et al. (2020); Zhang et al. (2020). *Parameter-isolation* protects models of old tasks using masks and/or network expansion Ostapenko et al. (2019); von Oswald et al. (2020); Li et al. (2019); Hung et al. (2019); Rajasegaran et al. (2020); Abati et al. (2020); Wortsman et al. (2020); Saha et al. (2021). Zhu et al. (2021a) used data augmentations to learn transferable features. Several papers are also based on orthogonal projection, e.g., OWM Zeng et al. (2019), OGD Farajtabar et al. (2020) and TRGP Lin et al. (2022). LwM Dhar et al. (2019) and RRR Ebrahimi et al. (2021) maintain the attention map of old tasks to tackle CF. They do not study the issue of learning invariant features. Thus, their feature map may learn variant features.

**Online CL methods** are mainly based on replay. ER randomly samples the replay data Chaudhry et al. (2020), MIR chooses replay data whose losses increase most Aljundi et al. (2019a), ASER uses the Shapley value theory Shim et al. (2021), and GDumb produces class balanced replay data Prabhu et al. (2020). GSS diversifies the gradients of the replay data Aljundi et al. (2019b). DER++ uses knowledge distillation Buzzega et al. (2020), SCR Mai et al. (2021b),Co2L Cha et al. (2021) and DualNet Pham et al. (2021) use self-supervised loss, and NCCL calibrates the network Yin et al. (2021). PASS Zhu et al. (2021b) uses rotation augmentation to create pseudo-classes. Bang et al. (2021) and Bang et al. (2022) proposed two blurry online CL settings.

**Data augmentations** have been used in many traditional learning and CL methods. Existing methods Buzzega et al. (2020); Pham et al. (2021); Mai et al. (2021a) use data augmentations (e.g., cropping, rotation, adding noise, etc) to diversify the data for better performance. They do not focus on diversifying the environments to learn invariant representations as we do. Data augmentations also create pseudo-classes Zhu et al. (2021a), which change the semantics of the original classes. PAR Zhang et al. (2022) uses augmentation repeatedly to help alleviate memory overfitting.

**Domain generalization** (DG) learns a model with inputs from multiple given source domains (or environments) with the same class labels and test with inputs from unseen domains. Many DG methods leverages data augmentations or auxiliary data to expand the diversity of the source domains Wang et al. (2020); Wu et al. (2020); Arjovsky et al. (2019); Rame et al. (2022); Yang et al. (2021); Yue et al. (2019). Our training data have no identified domains and DG does not do CL.

## 3 PROBLEM FORMULATION AND BACKGROUND

We consider three online CL settings. (1) **disjoint tasks setting**, where the system learns a sequence of tasks one after another. Each task consists of several classes. The data for each task comes in a stream and the learner sees the data only once. In a replay method, when a small batch of data of the current task $t$ arrives $D_t^{new} = (X_t^{new}, Y_t^{new})$ (where $X_t^{new}$ is a set of new samples and $Y_t^{new}$ is its set of corresponding labels), a small batch of replay data $D_t^{buf} = (X_t^{buf}, Y_t^{buf})$ is sampled from the memory buffer $\mathcal{M}$ and used to jointly train in one iteration. (2) **blurry task setting** Koh et al. (2021), where the data of classes from previous tasks may appear again later. (3) **data environment shift setting**, where the classes are fixed but the environments of classes change with time.

Our model $F$ consists of a feature extractor $f_\theta$ (where $\theta$ is the parameter set of the feature extractor) and a classifier $\sigma_\phi$ (where $\phi$ is the parameter set of the classifier). $f_\theta$ extracts features from the input $x$ to form a high-level representation, and $F(x) = \sigma_\phi(f_\theta(x))$ outputs the logits for each class.

# 4    THEORETICAL ANALYSIS OF INVARIANT REPRESENTATION LEARNING

In this section, we prove that learning invariant class representation is necessary for online CIL and also for general CIL. Our proof follows the data generation mechanism framework of the previous invariant feature learning works Arjovsky et al. (2019); Zhou et al. (2022). First, we formally describe the class learning in one task: In learning task $t$, we use $x_c^{Z_t}$ to denote the input of class $c$ of the task $t$ from the environment variable $Z_t$. $x_c^{Z_t}$ is the concatenation of invariant features $x_c^{inv} \in R^{d_{inv,c}}$ (generated by $Y_t$), variant features $x_{z_t}^{var} \in R^{d_{var}}$ (generated by $Z_t$), and random features $x^r \in R^{d_r}$ (random noise following a sub-Gaussian distribution with zero mean and bounded variance), i.e., $x_c^{Z_t} = [x_c^{inv}, x_{z_t}^{var}, x^r] \in R^d$. To learn the mapping relation between $x_c^{Z_t}$ and class $c$, we train a linear model $F(\cdot; \theta, \phi)$ with the cross-entropy loss $\mathcal{L}_{ce}(F(x_c^{Z_t}; \theta, \phi))$. Following the linear model assumption, the output $F(x_c^{Z_t}; \theta, \phi)$ for class $c$ is:

$$F(x_c^{Z_t}; \theta, \phi_c) = (\theta \circ x_c^{Z_t})^T \phi_c + b \tag{1}$$

where $\phi_c \in R^d$ is the classifier weights for class $c$ and $d$ is the input dimension. The parameter of feature extractor $\theta$ turns into a binary vector $\theta \in \{0, 1\}^{d \times 1}$ performing feature selection here and $b$ is the bias and $\circ$ is the element-wise product.

Next, we formally describe the learning of new tasks in online CIL and the forgetting of previous classes: When the next task $t + 1$ arrives, the class distribution changes and we denote the input of a new class $c'$ in task $t + 1$ as $x_{c'}^{Z_{t+1}}$. The linear model learns the prediction of class $c'$ based on $x_{c'}^{Z_{t+1}}$. For the previous class $c$, we define its forgetting after learning the new class $c'$ as:

$$\mathbb{H}(c, c') = \mathbb{E}_{x_c^{Z_t}}[-log(\frac{e^{F(x_c^{Z_t}; \theta_{t+1}, \phi_{c,t+1})}}{e^{F(x_c^{Z_t}; \theta_{t+1}, \phi_{c,t+1})} + ... + e^{F(x_c^{Z_t}; \theta_{t+1}, \phi_{c',t+1})}}) - (-log(\frac{e^{F(x_c^{Z_t}; \theta_t, \phi_{c,t})}}{e^{F(x_c^{Z_t}; \theta_t, \phi_{c,t})} + ...}))] \tag{2}$$

where $x_c^{Z_t}$ is an input sampled from class $c$'s distribution. The first term on the right is the expected CE loss on class $c$'s distribution after training task $t + 1$ that includes class $c'$. The second term is the expected CE loss on class $c$'s distribution after the training of task $t$ that includes class $c$ but before learning task $t + 1$. '...' is the sum of the terms for the logits of other classes. $\{\theta_t, \phi_{c,t}\}$ and $\{\theta_{t+1}, \phi_{c,t+1}\}$ are the parameters after training task $t$ and task $t + 1$, respectively.

Now we assume that (i) the model has converged to an optimal point after training task $t$. (ii) There exists an ideal CL algorithm that protects all important weights related to previous classes when learning task $t + 1$ like that in (Kim et al., 2022) using masks. We will show that the class representation involving variant features has higher forgetting even with the two ideal conditions.

When the environment variable $Z_t$ reappear in task $t + 1$ (i.e., $x_{z_t}^{var}$ are in the input of new task data), there exists three different situations and corresponding conclusions: (1) if the model $F$ only uses invariant features of class $c$ to form class representation ($\theta_t$ assigns 0 to $x_{z_t}^{var}$), then we have $\mathbb{H}(c, c')$ is independent of whether $F$ uses variant features in the class $c'$'s representation or not. (2) If the model $F$ uses both invariant features and variant features of class $c$ to form class representation, then $\mathbb{H}(c, c')$ is affected by whether $F$ uses variant features to form the class $c'$'s representation. We denote the forgetting of class $c$ for the cases without using variant features and with using variant features to form the class $c'$'s representation as $\mathbb{H}(c, \overline{c'})$ and $\mathbb{H}(c, \overline{\overline{c'}})$ respectively. We have $\mathbb{H}(c, \overline{c'}) > \mathbb{H}(c, \overline{\overline{c'}})$. (3) If the model $F$ only uses variant features to form class $c$'s representation,

then $\mathbb{H}(c, c')$ is affected by whether $F$ uses variant features to form the class $c'$'s representation. We have similar conclusions as (2). The proof is given in Appendix A.

Based on the analysis of three situations, we know that using variant features in the class representation can have the risk of causing higher forgetting when a class from some future tasks also uses the same variant features to form part of its class representation. In contrast, methods using only invariant features for class representations avoid this risk. Our findings show that learning variant features for class representation in a straightforward linear scenario risks increased forgetting, let alone in complex non-linear deep networks.

## 5  LEARNING INVARIANT FEATURES

In this section, we first present a general invariant feature learning optimization objective based on the above analysis. Then we propose environment augmentations to achieve our optimization objective in online CIL. We deploy our objective with augmentations into the basic experience replay approach and call the proposed technique IFO (**I**nvariant **F**eature learning for **O**nline CL). We justify that our method is the approximation of the Interventional Empirical Risk in online CIL.

### 5.1  INVARIANT FEATURE LEARNING OPTIMIZATION OBJECTIVE

Based on the above analysis in Sec. 4, our learning goal for the over-parameterized linear model to learn one class $c$ in task $t$ is (1) $\min_F \mathcal{L}_{ce}(F(x_c^{Z_t}; \theta, \phi))$ (well-trained model for classifying class $c$) while (2) maintaining $\theta^* = [1^{d_{inv,c}}, 0^{d_{var}}, 0^{d_r}]$ (only using invariant features to form the class representation). When we consider the more general case (e.g., a non-linear model and every class in task $t$), the latter goal (2) turns into $P(Z_t|f_\theta(x^{Z_t})) = P(Z_t)$ (the representation of each input $x$ of the class is independent of the environment variable).

To satisfy the second goal, we propose a new *alignment loss* here. Assume that we have a **training environment set** $\mathbb{Z}_t$ in the training data which has multiple different environments $z$ and we can control the environments in $\mathbb{Z}_t$. The *alignment loss* forces the representations of the images with the same class label but sampled from different environments in the new data batch $D_t^{new}$ to be close:

$$\mathcal{L}_{align}(D_t^{new}) = \frac{\sum_x \sum_{x'} \mathbb{I}(y = y' \wedge z \neq z') \cdot \mathbb{L}(f_\theta(x), f_\theta(x'))}{\sum_x \sum_{x'} \mathbb{I}(y = y' \wedge z \neq z')} \tag{3}$$

where $y/y'$ is the label of $x/x'$, and $z/z'$ is the environment that $x/x'$ sampled from. $\mathbb{L}(\cdot, \cdot)$ computes the cosine distance between the two representations and $\mathbb{I}$ is the indicator function. So the learned representation does not have any information to infer the environment variable. Then our invariant feature learning objective that considers (1) and (2) turns into:

$$\mathcal{L}_{IFO}(D_t^{new}) = \mathcal{L}_{ce}(D_t^{new}) + \mathcal{L}_{align}(D_t^{new}) \tag{4}$$

### 5.2  ENVIRONMENT AUGMENTATION BASED ON PRIOR KNOWLEDGE: METHOD I

The above section leaves an unsolved problem: How can one generate multiple different environments $z$ in online CIL? It's notoriously hard to automatically (i) identify all variant features in natural images and (ii) to adjust them to create different environments. Thus, in this section, we propose two simple environmental augmentations based on prior knowledge as the first attempt: (1) *color change*, which changes the color of the non-object part of the image as we know that the color of the non-object part usually does not contain the core invariant features of a class, and (2) *edge expansion*, which expands an image with the non-object edge part of another image as we know that the non-object edge part of an image is usually irrelevant to the classification task and can be treated as containing variant features. Note that we do not assume that the object of the class is centered in the image and these augmentations need to modify the environment while preserving the semantic meaning of the class. We introduce the details of the augmentations below.

**(1).** *Color Augmentation for the Non-object Part.* For an input image $x$, (1) we calculate the class activation map (CAM) Zhou et al. (2016) of $x$ that identifies the regions of $x$ attended to by a classifier. We set a mask of the object in $x$ as $M(x) = CAM(x) > \alpha$, where $\alpha$ is an importance threshold. (2) We then create another image $x'$ by randomly reordering the RGB channels of the

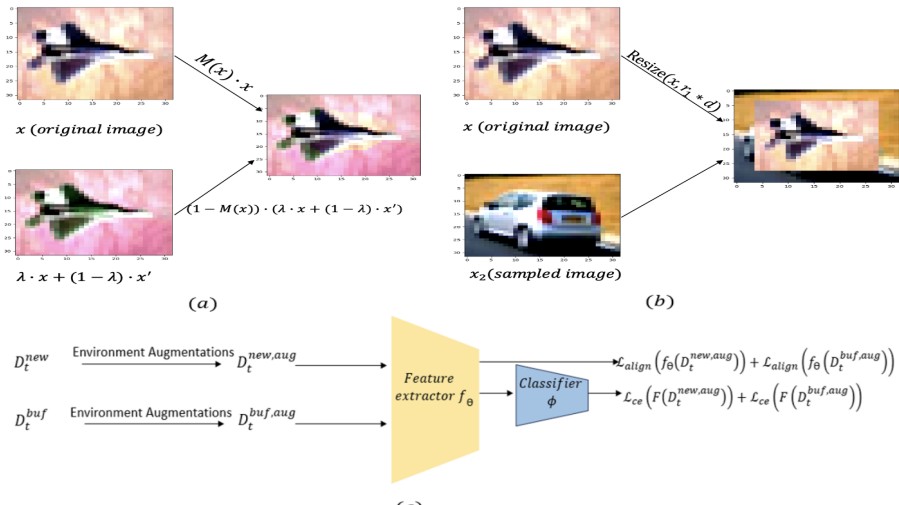

Figure 1: (a): Illustration for data augmentation $Aug_{color}$. (b): Illustration of data augmentation $Aug_{expand}$. (c): Illustration of the proposed invariant feature learning. (1) we first sample a new data batch $D_t^{new}$ and replay buffer data batch $D_t^{buf}$. (2) We create environment-augmented images from different environments. Then we combine them with the original data batch to construct the augmented data batches $D_t^{new,aug}$ and $D_t^{buf,aug}$. (3) We calculate the $\mathcal{L}_{ce}$ loss of the augmented data batches and the $\mathcal{L}_{align}$ loss by using their hidden representations.

original image $x$, and (3) sample an $\lambda$ value from the Beta distribution. The generation of the augmented image $\hat{x}$ is:

$$\hat{x} = (1 - M(x)) \cdot (\lambda \cdot x + (1 - \lambda) \cdot x') + M(x) \cdot x \tag{5}$$

In this augmentation, the original class semantic feature (e.g., shape) is protected by the object mask $M(x)$ and the color of the non-object part varies with different orders of the RGB channels and $\lambda$. We denote this augmentation as $Aug_{color}(x, s)$, where $s$ means that we repeat the augmentation $s$ times to create $s$ different augmented images. In section 6.4, we demonstrate that augmenting without the mask already yields strong results, and using the mask further improves the result.

**(2). *Edge Expansion Augmentation*.** In this augmentation, we expand $x$ by extracting the edge of another image $x'$ and using it to expand $x$. Note that the element in the original edge and center of $x$ is unchanged. Specifically, for input $x$ of size $n$, (1) we resize $x$ to $r_1 \cdot n$ where $r_1$ is the resize rate. We call this resized image $resize(x, r_1 \cdot n)$, which maintains the original class semantic features. (2) We randomly sample an image $x''$ from the batch $X_t^{buf} \bigcup X_t^{new}$ and replace the object part of $x''$ ($M(x'') \cdot x''$) with $resize(x, r_1 \cdot n)$. We use $M(x'')$ to identify the location of the object part rather than assume that it is in the center. If $resize(x, r_1 \cdot n)$ can not cover the whole object part of $x''$, we sample an image again. We denote this augmentation as $Aug_{expand}(x)$. We illustrate the two augmentations in Figure 1 (a) and (b).

### 5.3 Environment Augmentation via Clustering the Replay Data: Method II

This section, propose the second environmental augmentation method based on the property of the online CIL setting. In online CIL, earlier training data batches of a task may have very different environments from the subsequent training data batches. The shift of environments in one task means previous training batches can be utilized to augment the environments of the newest training batch. However, the model cannot revisit previous batches. So we use the stored data for the current task in the replay buffer as it is updated continuously. Specifically, (1) for each class of the current task, we use the $k$-means method to cluster the representations of the data of the class in the buffer into $k$ clusters (pseudo environments). (If the number of stored samples for a new class is smaller than $k$, we do not cluster this class's data. (2) For each image $x$ in $D_t^{new}$, we collect the clusters that have the same class as $x$ and calculate the cosine distance of $f_\theta(x)$ and each cluster's prototype. (3) We search the cluster that has the minimum distance to $f_\theta(x)$ and view $x$ as a sample from the environment of this cluster. (4) We sample one data sample $x_p$ per cluster from the other clusters of

the same class and view $x_p$ as the sample from another environment. If the number of stored samples for the class of $x$ is smaller than $k$, we choose the same class's stored data sample that has the maximum representation cosine distance to $x$ as $x_p$. (5) We use $x_p$ as $x''$ in the above augmentation $Aug_{expand}(x)$ to create more environments and denote $Aug_{expand}(x)$ using $x_p$ as $Aug_{expand}^p(x)$. This method augments new data $x$ with $k-1$ previously seen but different environments of the same class via replaying the current task's data, i.e., training it makes the model learn an invariant class representation that generalizes well for the same class's data points sampled from different times.

## 5.4 PUT EVERYTHING TOGETHER

After augmenting each image $x$ in the new data batch, we combine all augmented images with the original images to construct the augmented new data batch $D_t^{new,aug}$. We then calculate the invariant loss $\mathcal{L}_{IFO}(D_t^{new,aug})$. The **replay data** also needs to learn invariant representations as well to avoid over-fitting the few samples in the buffer. So we also augment the replay data batch as $D_t^{buf,aug}$ and calculate $\mathcal{L}_{IFO}(D_t^{buf,aug})$. The final invariant loss for the IFO method (Figure 1 (c)) is:

$$\mathcal{L}_{IFO}(D_t^{new,aug}) + \mathcal{L}_{IFO}(D_t^{buf,aug}) = \mathcal{L}_{ce}(D_t^{new,aug}) + \mathcal{L}_{align}(D_t^{new,aug}) + \mathcal{L}_{ce}(D_t^{buf,aug}) + \mathcal{L}_{align}(D_t^{buf,aug}) \quad (6)$$

**Relationship with Causal Inference.** Learning invariant features naturally follows the causal inference theory because invariant features are the real causes for the class. In the causal inference theory, Jung et al. (2020) showed that we need to replace the *observational distribution* $P(Y_t|X_t)$ with the *interventional distribution* $P(Y_t|do(X_t))$ in *Empirical Risk Minimization (ERM)* for learning causal features, where $do(X)$ removes the environmental effects from the prediction of $Y$. Then Wang et al. (2022b) proposed the *Interventional Empirical Risk* to learn casual features that are invariant to **the ideal environment variable** $Z$ that includes all possible environments:

$$\begin{aligned} \hat{R}_Z(D_t) &= \mathbb{E}_{x \sim P(X_t), y \sim P(Y_t|do(X_t))}\mathcal{L}_{ce}(F(x), y) \\ &= \sum_y \sum_x \sum_{z \in Z} \mathcal{L}_{ce}(F(x), y)P(y|F(x), z)P(z)P(F(x)) \end{aligned} \quad (7)$$

where $z$ is a sample environment from the ideal environment set $Z$ and $D_t$ is a data batch of task $t$ (e.g., new data batch or replay data batch). Optimizing this loss is thus equivalent to learning invariant features as variant features from different environments are filtered out by $do(X)$.

We prove here that with the guarantee of optimizing the alignment loss $\mathcal{L}_{align}$ (eq. 3), optimizing the CE loss of $D_t$ is an approximation of the *Interventional Empirical Risk* of $D_t$ based on the training environment set $\mathbb{Z}$. And Eq. 6 is the approximation of the *Interventional Empirical Risk* of $D_t^{new,aug} + D_t^{buf,aug}$. With the alignment loss, the model learns little variant environment information about the training environment set $\mathbb{Z}$, which means $P(\mathbb{Z}|f_\theta(x)) \approx P(\mathbb{Z})$ (the mutual information $I(\mathbb{Z}; f_\theta(x)) \approx 0$). According to the Data Processing Inequality, we can infer that $I(\mathbb{Z}; f_\theta(x)) \geq I(\mathbb{Z}; F(x)) \geq 0$ as the environment information in the input decreases layer by layer. Then with the alignment loss, $I(\mathbb{Z}; F(x)) \approx 0$ ($P(\mathbb{Z}|F(x)) \approx P(\mathbb{Z})$) and we have:

$$\begin{aligned} \mathcal{L}_{ce}(D_t) &= \sum_y \sum_x \sum_z \mathcal{L}_{ce}(F(x), y)P(F(x), y, z) \\ &\approx \sum_y \sum_x \sum_{z \in \mathbb{Z}} \mathcal{L}_{ce}(F(x), y)P(y|F(x), z)P(z)P(F(x)) \end{aligned} \quad (8)$$

The two environment augmentations are for approximating the *Interventional Empirical Risk* based on the ideal set $Z$ (diversifying the empirical environment set $\mathbb{Z}$ to make it closer to the ideal set $Z$).

## 6 EXPERIMENT RESULTS

We evaluate the proposed method in three online CL scenarios: standard *disjoint task* scenario, *blurry task boundary* scenario, and *data environment shift* scenario.

### 6.1 DISJOINT ONLINE CONTINUAL LEARNING SCENARIO

In this scenario, the tasks come one after another and each task have its unique classes.

**Datasets and Baselines.** we use five popular image classification datasets. For MNIST LeCun et al. (1998), we split its 10 classes into 5 different tasks, 2 classes per task. For CIFAR10 Krizhevsky & Hinton (2009), we also split its 10 classes into 5 different tasks with 2 classes per task. For CIFAR100 Krizhevsky & Hinton (2009), we split its 100 classes into 10 different tasks with 10 classes per task. For TinyImagenet Le & Yang (2015), we split its 200 classes into 100 different tasks with 2 classes per task for stress testing. For ImageNet Deng et al. (2009), we split its 1000 classes into 10 different tasks with 100 classes per task. Each task runs with only one epoch for online CL. We use 9 online CL baselines shown in Table 1. We run their official codes (Appendix B).

**Backbone, Optimizer, Batch size.** We follow Guo et al. (2022) and use ResNet-18 (*not pre-trained*) as the backbone for all methods in the CIFAR10, CIFAR100, TinyImageNet, and ImageNet settings. A fully connected network with two hidden layers (400 ReLU units) is used as the backbone for MNIST. We use the Adam optimizer and set the learning rate as 1e-3 for our method and set the new data batch size $N^{new}$ and buffer data batch size $N^{buf}$ as 10 and 64 respectively for all methods and use the reservoir sampling for our method. We train each training batch with one iteration.

**Data Augmentation.** Following Guo et al. (2022), the data augmentation methods *horizontal-flip*, *random-resized-crop*, and *random-gray-scale* are applied to all methods (except methods with their specific augmentations) to improve the performance (no method has experienced performance drop). We set $\alpha$ in the object mask as 0.25 by grid search and $s$ in $Aug_{color}(x, s)$ as 5. We set $r_1$ in $Aug_{add}$ as 0.75 and set the number of clusters $k$ for each class as 3. More details are given in Appendix C.

| Method | MNIST | | | CIFAR10 | | | CIFAR100 | | | TinyIN | | |
|---|---|---|---|---|---|---|---|---|---|---|---|---|
| B | B=0.1k | B=0.5k | B=1k | B=0.2k | B=0.5k | B=1k | B=1k | B=2k | B=5k | B=2k | B=4k | B=10k |
| AGEM Chaudhry et al. (2018) | 56.9±5.2 | 57.7±8.8 | 61.6±3.2 | 22.7±1.8 | 22.7±1.9 | 22.6±0.7 | 5.8±0.2 | 5.8±0.3 | 6.5±0.2 | 0.9±0.1 | 2.1±0.1 | 3.9±0.2 |
| GSS Aljundi et al. (2019b) | 70.4±1.5 | 80.7±5.8 | 87.5±5.9 | 26.9±1.2 | 30.7±1.3 | 40.1±1.4 | 11.1±0.2 | 13.3±0.5 | 17.4±0.1 | 3.3±0.5 | 10.0±0.2 | 10.5±0.2 |
| ER Chaudhry et al. (2020) | 78.7±0.4 | 88.0±0.2 | 90.3±0.1 | 29.7±1.0 | 35.2±0.3 | 44.3±0.4 | 11.7±0.3 | 15.0±0.9 | 14.4±0.9 | 5.6±0.5 | 10.1±0.7 | 11.7±0.2 |
| MIR Aljundi et al. (2019a) | 79.0±0.5 | 88.3±0.1 | 91.3±1.9 | 37.3±0.3 | 40.0±0.6 | 41.0±0.6 | 15.7±0.2 | 19.1±0.1 | 24.1±0.2 | 6.1±0.5 | 11.7±0.2 | 13.5±0.2 |
| ASER Shim et al. (2021) | 61.6±2.1 | 71.0±0.6 | 82.1±5.9 | 27.8±1.0 | 36.2±1.2 | 44.7±1.2 | 16.4±0.3 | 12.2±1.9 | 27.1±0.3 | 5.3±0.3 | 8.2±0.2 | 10.3±0.4 |
| GDumb Prabhu et al. (2020) | 81.2±0.5 | 91.0±0.2 | 94.5±0.1 | 35.9±1.1 | 50.7±0.7 | 63.5±0.5 | 14.1±0.3 | 20.1±0.2 | 36.0±0.5 | 12.6±0.1 | 12.7±0.3 | 15.7±0.2 |
| DualNet Pham et al. (2021) | 85.2±0.4 | 91.5±0.2 | 94.2±0.2 | 45.3±1.2 | 52.6±0.7 | 60.2±0.5 | 17.2±0.6 | 27.5±1.6 | 31.5±0.5 | 10.3±0.2 | 18.2±0.3 | 20.3±0.2 |
| SCR Mai et al. (2021b) | 86.2±0.5 | 92.8±0.3 | 94.6±0.1 | 47.2±1.7 | 58.2±0.5 | 64.1±1.2 | 26.5±0.2 | 31.6±0.5 | 36.5±0.2 | 10.6±1.1 | 17.2±0.1 | 20.4±1.1 |
| OCM Guo et al. (2022) | 90.7±0.1 | 95.7±0.3 | 96.7±0.1 | 59.4±0.2 | 70.0±1.3 | 77.2±0.5 | 28.1±0.3 | 35.0±0.4 | 42.4±0.5 | 15.7±0.2 | 21.2±0.4 | 27.0±0.3 |
| IFO (ours) | 86.8±0.2 | 94.1±0.1 | 94.8±0.4 | 46.0±0.5 | 56.7±0.4 | 63.6±0.4 | 34.7±0.2 | 38.1±0.5 | 46.6±0.3 | 13.5±0.5 | 20.5±0.4 | 28.5±0.3 |
| IFO+OCM (ours) | 92.5±0.4 | 96.1±0.2 | 97.0±0.2 | 65.0±0.3 | 73.2±0.2 | 78.0±0.3 | 38.5±0.3 | 47.3±0.1 | 53.1±0.3 | 21.6±0.5 | 26.2±0.3 | 34.5±0.4 |

Table 1: Accuracy on the MNIST (5 tasks), CIFAR10 (5 tasks), CIFAR100 (10 tasks), and TinyIN (TinyImageNet 100 tasks) datasets with different memory buffer sizes $B$. All values are the averages of 15 runs. See the results on ImageNet in Figure 2(a).

**Accuracy results.** We report the average accuracy of all tasks *after learning the final task* in Table 1. We observe that our IFO improves the performance of ER by a large margin and also outperforms almost all baselines. Further, the combined method IFO+OCM performs significantly better than all baselines including OCM as IFO+OCM learns both *invariant* and *holistic representations*. IFO+OCM is especially strong when the buffer size $B$ is small (e.g., 10 samples per class), as the baselines tend to over-fit the buffer data and learn variant features when $B$ is small. IFO+OCM can avoid this by learning invariant and holistic features. Our IFO+OCM also outperforms four batch CL baselines in the online CL setting (see Table 4 in Appendix C).

For the ImageNet dataset, due to the poor overall performance, we compare IFO+OCM with top 3 baselines (OCM, SSIL, and SCR). Figure 2(a) shows the accuracy of all tasks seen so far after learning each task. IFO+OCM consistently outperforms the three baselines in the whole process. The accuracy first arises and then drops as the random-initialized model doesn't have enough features to solve the first task until the second task arrives. The later drop is due to CF.

**Forgetting rate and efficiency analysis.** We report the average forgetting rate Chaudhry et al. (2020) in Table 5 in Appendix C. As a replay method, IFO drops the forgetting rate significantly compared to the original ER, which shows IFO's ability to avoid forgetting by reducing the shared irrelevant features. Further, IFO+OCM forgets the least except for DualNet, GDumb, and SCR on TinyImageNet. But the accuracy of the three baselines is much lower than IFO+OCM. For ImageNet, IFO+OCM also fares well. See Appendix C on **time efficiency comparison**.

**Learning invariant representations.** Here we want to use two types of metrics to quantitatively show that our IFO has indeed learned more invariant features: **(1)** representation similarity of the same class's data across different environments. **(2)** model robustness on unseen environments.

We separately train a ResNet-18 model using cross-entropy loss and a model using IFO loss (Eq. 4) on the original CIFAR10, CIFAR100, and Tiny-ImageNet datasets in one epoch respectively. Then

| Loss | CIFAR10-C | CIFAR100-C | Tiny-ImageNet-C |
|---|---|---|---|
| Cross-entropy loss | 89.0±0.2 | 87.9±0.1 | 85.7±0.3 |
| Our $\mathcal{L}_{IFO}$ (Eq. 4) | 97.0±0.2 | 95.6±0.3 | 94.8±0.3 |

Table 2: Representation CKA correlation across different environments - the average of 5 runs.

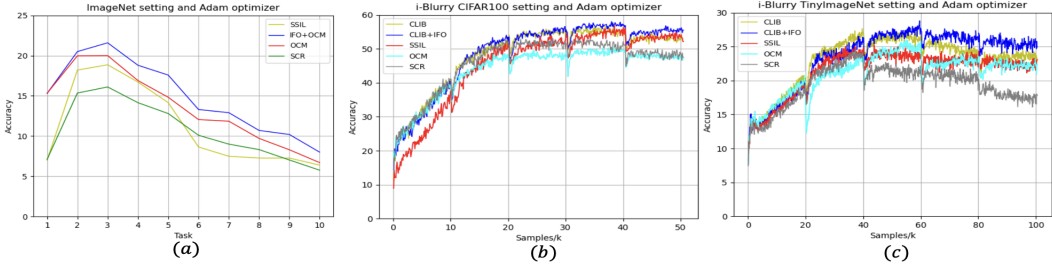

Figure 2: (a) results in the disjoint setting using ImageNet. The buffer size is 10k. For the online i-Blurry setting in (b) and (c), 'samples' of the x-axis means the number of new data samples that the model has seen and the unit is 1k. (b) results of the CIFAR100 dataset. The buffer size is 2k. (c) results of the TinyImageNet dataset. The buffer size is 4k.

| Dataset | no $Aug_{color}$ | no $Aug_{add}$ | no $Aug_{add}^p$ | no align | no new data | no mask | IFO+Mixup | IFO+CutMix | IFO+ClassAug | IFO+ClassAug | IFO+RAR | IFO+LwM | IFO+RRR |
|---|---|---|---|---|---|---|---|---|---|---|---|---|---|
| CIFAR100 | 35.1±0.7 | 37.4±0.5 | 37.0±0.2 | 36.2±0.2 | 35.0±0.1 | 37.3±0.1 | 24.0±0.5 | 27.1±0.3 | 26.5±0.1 | 30.3±0.4 | 30.9±0.2 | 39.0±0.2 | 40.1±0.3 |
| TinyImageNet | 12.0±0.2 | 12.2±0.2 | 12.7±0.2 | 12.8±0.2 | 11.9±0.5 | 13.0±0.2 | 6.3±0.2 | 7.9±0.5 | 9.4±0.3 | 9.6±0.5 | 9.9±0.3 | 13.6±0.3 | 14.9±0.5 |

Table 3: Ablation accuracy - an average of 5 runs. The memory buffer size is 2k.

we use the CIFAR10-C, CIFAR100-C, and Tiny-ImageNet-C datasets Hendrycks & Dietterich (2019a) (which contain 19 different corrupted environments) to calculate the average CKA (*centered kernel alignment*) correlation Kornblith et al. (2019) of the hidden representations from the trained model across different environments. Specifically, we calculate the correlation of the hidden representations of each class in each pair of environments $z$ and $z'$ as $CKA(f_\theta(X_c^z), f_\theta(X_c^{z'}))$, where *CKA* is the CKA correlation metric, and then calculate the average of $CKA(f_\theta(X_c^z), f_\theta(X_c^{z'}))$ over all possible pairs of environments. We denote it as $Avg(CKA(f_\theta(X_c^z), f_\theta(X_c^{z'})))$. Finally, we compute the mean $Avg(CKA(f_\theta(X_c^z), f_\theta(X_c^{z'})))$ across all classes. From table 2, we see that IFO has learned class representations that have high similarities across different environments (high invariance). For (2), we show that our IFO generalizes well to unseen environments (see Appendix D).

## 6.2 BLURRY ONLINE CONTINUAL LEARNING SCENARIO

In this scenario Koh et al. (2021), the classes of the training data for a dataset are split into two parts, $N\%$ of the classes as the disjoint part and the rest of $100 - N\%$ of the classes as the blurry part. Each task consists of some disjoint classes (with all their data) and some blurry classes. The disjoint classes of a task do not appear in any other task. For the blurry class data in a task, $100 - M\%$ of the samples are from some randomly selected dominant blurry classes (these classes will not be dominant classes in other tasks) and $M\%$ of the samples are from the other minor classes.

Following Koh et al. (2021), we use the i-Blurry setup with $N = 50$ and $M = 10$ (i-Blurry-50-10) in our experiments on the CIFAR100 (5 tasks) and TinyImageNet (5 tasks) datasets. We use ResNet-34 and Adam optimizer with an initial learning rate of 0.0003 for all systems. We compare their original system CLIB with our CLIB+IFO (replacing CLIB's cross-entropy loss with our $\mathcal{L}_{IFO}$ loss in Eq. 4) and three best-performing baselines (OCM, SSIL, and SCR). More details are in Appendix E.

**Accuracy results.** Following Koh et al. (2021), we measure any time inference and plot the accuracy-to-samples curve in Figure 2 (b)&(c). The setting is as follows: After the model sees every 100 new samples, we test the model using the original test set of all classes in the CIFAR100 or TinyImageNet dataset and record the accuracy. The figures show that CLIB+IFO outperforms CLIB, especially after the model has seen some tasks, which indicates that learning invariant representations improves the overall generalization ability. We also observe that the three best baselines' performances are weaker than that of CLIB as the three baselines are not designed for this setting.

### 6.3 Environment Shift in Online Continual Learning

In this scenario, the model learns the same classes from different environments sequentially and is then tested in an unseen environment. We use the PACS data Li et al. (2017) to simulate this scenario as it has four different environments. More details are in Appendix G. Table 7 in Appendix F shows that the performance of our $\mathcal{L}_{IFO}$ loss (Eq. 4) is beyond that of the cross-entropy loss $\mathcal{L}_{ce}$ as $\mathcal{L}_{IFO}$ loss learns invariant features better. Another observation is that optimizing the classification loss ($\mathcal{L}_{IFO}$ or $\mathcal{L}_{ce}$) with the $\mathcal{L}_{OCM}$ loss improves the performance further as learning holistic features of one class makes the model have more knowledge to deal with samples from unseen environments.

### 6.4 Ablation Study and Analysis

Here we use the traditional disjoint online CL scenario.

**Ablation study of the components in IFO** Table 3 shows the results without using $Aug_{color}$ (no $Aug_{color}$) or $Aug_{add}$ (no $Aug_{add}$). Their results are weaker, which shows the contribution of the proposed augmentations. For "no $Aug_{add}^p$", we replace $Aug_{add}^p$ with $Aug_{add}$ in augmenting new data batch. The result becomes worse as the pseudo environments in the stored new task data are not used. For "no align", we optimize the model with each created environment without $\mathcal{L}_{align}$ of Eq. 6 and the performance is poorer as the model tends to memorize each environment and does not learn invariant features. For "no new data", only $D_t^{buf}$ in Eq. 6 is used. The performance also drops, meaning that considering new samples to learn invariant features is useful. For 'no mask', we set $\hat{x} = \lambda \cdot x + (1 - \lambda) \cdot x'$ for $Aug_{color}$ and the performance is worse as the object color is an important feature and should not be changed randomly.

**Other data augmentations**. We tried some popular data augmentations for feature learning to create environments for IFO: Mixup Zhang et al. (2017) (IFO+Mixup)), Cutmix Yun et al. (2019) (IFO+CutMix), MemoryAug Fini et al. (2020) (IFO+MemoryAug), ClassAug Zhu et al. (2021a) (IFO+ClassAug) and RAR Zhang et al. (2022) (IFO+RAR). However, the results of these augmentations (Table 3) are poorer than that of IFO with our environment augmentations as IFO needs to learn invariant class representations by only changing variant features.

**Maintaining salient feature maps.** This approach in LwM Dhar et al. (2019) and RRR Ebrahimi et al. (2021) for dealing with CF is complementary to IFO. IFO+LwM and IFO+RRR (Table 3) perform slightly better than IFO, but they are much weaker than IFC+OCM (Table 1).

**Influence of hyperparameters in IFO**. For $s$ in augmentation $Aug_{color}$ (Sec. 5.2), from Table 8 in Appendix G, we observe that the performance has a positive correlation with the number $s$ as the model gradually focuses on invariant features rather than strongly depending on simple colors. We set $s$ to 5 as it achieves the best result. For rate $r_1$ in the augmentation $Aug_{add}$, we need to avoid introducing trivial features ($r_1$ is too high) and causing a huge information loss in the original image ($r_1$ is too low). Based on Table 8 in Appendix H, we set $r_1$ as 0.75. For the number of clusters $k$, we need to introduce more pseudo environments but the number of buffer data is limited (e.g., 10 samples per class). Based on Table 9 in Appendix H, we set $k$ as 3.

**Influence of learning invariant features for CL.** Comparing IFO+OCM with OCM (see Figure 3 in Appendix H ), we find that IFO helps establish better class boundaries across tasks as IFO reduces the difficulty of establishing cross-task class boundaries by avoiding learning variant features.

## 7 Conclusion

Although many techniques have been proposed to solve class-incremental learning (CIL), limited work has been done to study the *necessary conditions* for feature learning to achieve good CIL performances. Guo et al. (2022) showed that it is necessary to learn *holistic features*. This paper demonstrated that it is also necessary to learn *invariant features*, and proposed a novel method to do so. Experimental results showed that the new condition gave another major boost to online CIL performances. ***Limitations*** of this work are discussed in Appendix I.

ETHICS

Because our research focuses solely on classification learning using publicly available datasets, and our algorithms aim to address the broader challenge of online continual learning rather than solving any specific application problems, we believe there are no ethical concerns associated with this study. We also do not foresee any negative societal consequences stemming from our approach.

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

## A    PROOF FOR THE PROPOSITION

**Proof:** Based on the assumption (i) that the model has converged to an optimal point after training task $t$, the model can classify class $c$ and other seen classes well. That means the sum of the terms for the logits of other classes (i.e., '...' in Eq. 2) is very small given the class $c$'s sample (e.g., In CIL, the assumption means the expected loss of class $c$ is 0 and then the sum is 0.). For simplicity, we delete this term in the following analysis. Based on assumption (ii), there exists an ideal CL algorithm that protects all important weights related to previous classes when learning task $t + 1$, we have $\theta_t^c \approx \theta_{t+1}^c$ and $\phi_{c,t} \approx \phi_{c,t+1}$ where $\theta^c$ and $\phi_c$ are the parameters in $\theta$ and $\phi$ that are related to class $c$.

Then we can turn the computation of the forgetting to:

$$\mathbb{H}(c, c') = \mathbb{E}_{x_c}[-log(\frac{e^{F(x_c^{Z_t};\theta_t,\phi_{c,t})}}{e^{F(x_c^{Z_t};\theta_t,\phi_{c,t})} + e^{F(x_c^{Z_t};\theta_t,\phi_{c',t+1})}})] \tag{9}$$

The proof of the conclusion in the situation (1) is obvious as the variant features in class $c$'s data are filtered out by $\theta_t^c$.

The proof of the conclusion in situation (2): To compare the forgetting of class $c$ for the case without using variant features to form the class $c'$'s representation $\mathbb{H}(c, \overline{c'})$ and the case with using variant features $\mathbb{H}(c, \overline{\overline{c'}})$, we can calculate their gap and simplify it:

$$\mathbb{H}(c, \overline{\overline{c'}}) - \mathbb{H}(c, \overline{c'}) = \log(\frac{e^{F(x_c^{Z_t};\theta_t,\phi_{c,t})}}{e^{F(x_c^{Z_t};\theta_t,\phi_{c,t})} + e^{F(x_c^{Z_t};\theta_t,\overline{\phi_{c',t+1}})}} \cdot \frac{e^{F(x_c^{Z_t};\theta_t,\phi_{c,t})} + e^{F(x_c^{Z_t};\theta_t,\overline{\overline{\phi_{c',t+1}}})}}{e^{F(x_c^{Z_t};\theta_t,\phi_{c,t})}})$$

$$= \log(\frac{e^{F(x_c^{Z_t};\theta_t,\phi_{c,t})} + e^{F(x_c^{Z_t};\theta_t,\overline{\overline{\phi_{c',t+1}}})}}{e^{F(x_c^{Z_t};\theta_t,\phi_{c,t})} + e^{F(x_c^{Z_t};\theta_t,\overline{\phi_{c',t+1}})}})$$

$$\tag{10}$$

where $\overline{\overline{c'}}$ means that the learned representation of class $c'$ involves variant features $x_{z_t}^{var}$ from the previous task.

To prove our conclusion that the class representation using variant features has a higher forgetting than the representation without using variant features, based on Eq.10, now the key is to prove $F(x_c^{Z_t};\theta_t,\overline{\overline{\phi_{c',t+1}}}) > F(x_c^{Z_t};\theta_t,\overline{\phi_{c',t+1}})$ (the last term in the numerator of Eq..10 is bigger than the last term in the denominator). Based on Eq. 1, we simplify the last term in the numerator of Eq..10 as $F(x_c^{Z_t};\theta_t,\overline{\overline{\phi_{c',t+1}}}) = (\theta_t \circ [x_c^{inv}, x_{z_t}^{var}, 0^r])^T \phi_{c',t+1} + b = F(x_c^{Z_t};\theta_t,\overline{\phi_{c',t+1}}) + (\theta_t \circ [0^{inv}, x_{z_t}^{var}, 0^r])^T \phi_{c',t+1}$. So the last term in the numerator of Eq.10 contains the last term in the denominator. We also have $(\theta_t \circ [0^{inv}, x_{z_t}^{var}, 0^r])^T \phi_{c',t+1} > 0$ as $\phi_{c',t+1}$ uses the variant feature $x_{z_t}^{var}$ to form class $c'$ representation. Then we have $F(x_c^{Z_t};\theta_t,\overline{\overline{\phi_{c',t+1}}}) > F(x_c^{Z_t};\theta_t,\overline{\phi_{c',t+1}})$ and their ratio is bigger than 1. Then we have $\mathbb{H}(c, \overline{\overline{c'}}) - \mathbb{H}(c, \overline{c'}) > log(1) = 0$, which proves our conclusion. The proof of the conclusion in situation (3) is similar to the above proof.

## B    ADDITIONAL DETAILS OF DISJOINT ONLINE CL SCENARIO

Due to the limitation of computational resources, we download the downsampled version of ImageNet ($3 \times 32 \times 32$) from the official website and conduct experiments on this dataset. $Aug_{color}$ is

not used on the MNIST dataset as it contains only black and white images. We set the $\alpha$ and the $\beta$ values of the $\beta$ distribution in $Aug_{color}$ as 1. To reduce computation, we cluster and assign the stored new class data into different clusters every 10 updates. If we do the clustering operation at a lower frequency, the representation knowledge learned in the model may shift and so we cannot capture it accurately.

For AGEM Chaudhry et al. (2018), following the original paper, we use random sampling to update the replay buffer and to sample data from the buffer.

For GSS Aljundi et al. (2019b), based on the original paper, we use the same optimizer and learning rate as above. The number of buffer batches randomly sampled from the memory to estimate the maximal gradients of the cosine similarity score is set to 10 and the randomly sampled buffer batch ($X^{buffer}$) size for calculating the score is 64.

For DualNet Pham et al. (2021), based on the original paper, we use the Look-ahead optimizer to train DualNet's slow learner and the memory is implemented as a reservoir buffer. Its architecture is the full ResNet-18. We follow it and train the slow learner with $n = 3$ iterations using the episodic memory data before observing a mini-batch of labeled data. We follow its official setting and only use the random cropping and flipping for the supervised training phase. For other optimizers and hyper-parameters, we follow the official code.

For ASER Shim et al. (2021), we follow the original paper and use the mean value of Adversarial SV and Cooperative SV, and set the maximum number of samples per class for random sampling as 1.5. We allow 3 nearest neighbors for KNN-SV computation. We use the same SV-based methods for both Memory-Update and Memory-Retrieval as given in the original paper.

For MIR/ER Aljundi et al. (2019a), we set the sub-sample size to 128 and follow the original paper to set other hyper-parameter.

For DER++ Buzzega et al. (2020), we follow the original paper and set the value of alpha ($\alpha$) as 0.1, and fix the beta ($\beta$) as 0.5.

For GDumb Prabhu et al. (2020), we follow the official code and use CutMix as the regularization to overcome over-fitting. we follow the official code and set the number of epochs for training the whole buffer data as 256 for MINIST, CIFAR10, and CIFAR100 datasets, and 32 for the TinyImagenet dataset. We set the gradient clip as 10.

For SCR Mai et al. (2021b), we follow the orginal paper and set the temperature for contrastive loss as 0.07. We employ a linear layer with the size [$dim_h$,128] as the contrastive head. We follow the official code and use the horizontal-flip, random-resized crop, random-gray-scale, and color-jitter as data augmentations.

For methods AGEM, ASER, MIR, ER, DER++ GDumb, and SCR, we follow Guo et al. (2022) and use the Adam optimizer and set the learning rate as 1e-3 to optimize them. For DualNet, it has specific optimization algorithms for fast and slow learners respectively, so we do not change it.

The official code of these systems can be found in the following locations.

ER and MIR: `https://github.com/optimass/Maximally_Interfered_Retrieval`.
DualNet: `https://github.com/phquang/DualNet`.
ASER and SCR: `https://github.com/RaptorMai/online-continual-learning`.
GDumb: `https://github.com/drimpossible/GDumb`.
DER++: `https://github.com/aimagelab/mammoth`.
AGEM: `https://github.com/facebookresearch/agem`.
GSS: `https://github.com/rahafaljundi/Gradient-based-Sample-Selection`.
Co$^2$L: `https://github.com/chaht01/Co2L`.
The code for IL2A: `https://github.com/Impression2805/IL2A`.
SSIL: `https://github.com/hongjoon0805/SS-IL-Official`.
OCM: `https://github.com/gydpku/OCM`.

Table 4: Accuracy on the MNIST (5 tasks), CIFAR10 (5 tasks), CIFAR100 (10 tasks), and TinyImageNet (100 tasks) datasets with different memory buffer sizes $B$. All values are the averages of 15 runs.

| Method | MNIST | | | CIFAR10 | | | CIFAR100 | | | TinyImageNet | | |
|---|---|---|---|---|---|---|---|---|---|---|---|---|
| **B** | B=0.1k | B=0.5k | B=1k | B=0.2k | B=0.5k | B=1k | B=1k | B=2k | B=5k | B=2k | B=4k | B=10k |
| DER++ Buzzega et al. (2020) | 74.4±1.1 | 91.5±0.2 | 92.1±0.2 | 44.2±1.1 | 47.9±1.5 | 54.7±2.2 | 15.3±0.2 | 19.7±1.5 | 27.0±0.7 | 4.5±0.3 | 10.1±0.3 | 17.6±0.5 |
| IL2A Zhu et al. (2021a) | 90.2±0.1 | 92.7±0.1 | 93.9±0.1 | 54.7±0.5 | 56.0±0.4 | 58.2±1.2 | 18.2±1.2 | 19.7±0.5 | 22.4±0.2 | 5.5±0.7 | 8.1±1.2 | 11.6±0.4 |
| Co$^2$L Cha et al. (2021) | 83.1±0.1 | 91.5±0.1 | 94.7±0.1 | 42.1±1.2 | 51.0±0.7 | 58.8±0.4 | 17.1±0.4 | 24.2±0.2 | 32.2±0.5 | 10.1±0.2 | 15.8±0.4 | 22.5±1.2 |
| SSIL Ahn et al. (2021) | 88.2±0.1 | 93.0±0.2 | 95.1±0.1 | 49.5±0.2 | 59.2±0.4 | 64.0±0.5 | 26.0±0.1 | 33.1±0.5 | 39.5±0.4 | 9.6±0.7 | 15.2±1.5 | 21.1±0.1 |
| IFO+OCM | **92.5**±0.4 | **96.1**±0.2 | **97.0**±0.2 | **65.0**±0.3 | **73.2**±0.2 | **78.0**±0.3 | **38.5**±0.3 | **47.3**±0.1 | **53.1**±0.3 | **21.6**±0.5 | **26.2**±0.3 | **34.5**±0.4 |

Table 5: Average forgetting rate. The table includes both the online CL baselines and the adapted online CL baselines from the 4 batch/offline CL systems. All numbers are the averages of 15 runs. See the forgetting rates for the ImageNet dataset in the text below.

| Method | MNIST | | | CIFAR10 | | | CIFAR100 | | | TinyImageNet | | |
|---|---|---|---|---|---|---|---|---|---|---|---|---|
| $B$ | B=0.1k | B=0.5k | B=1k | B=0.2k | B=0.5k | B=1k | B=1k | B=2k | B=5k | B=2k | B=4k | B=10k |
| AGEM Chaudhry et al. (2018) | 32.5±5.9 | 30.1±4.2 | 32.0±2.9 | 36.1±3.8 | 43.2±4.3 | 48.1±3.4 | 43.3±0.2 | 45.7±0.3 | 43.9±0.2 | 73.9±0.2 | 78.9±0.2 | 74.1±0.3 |
| GSS Aljundi et al. (2019b) | 26.1±2.2 | 17.8±5.22 | 10.5±6.7 | 75.5±1.5 | 65.9±1.6 | 54.9±2.0 | 30.8±0.2 | 30.7±0.5 | 26.4±0.3 | 72.8±1.2 | 72.6±0.4 | 71.5±0.2 |
| ER Chaudhry et al. (2020) | 22.7±0.5 | 9.7±0.4 | 6.7±0.5 | 42.0±0.3 | 26.7±0.7 | 20.7±0.7 | 34.2±0.2 | 31.7±0.9 | 35.3±0.9 | 68.2±2.8 | 66.2±0.8 | 67.2±0.2 |
| MIR Aljundi et al. (2019a) | 22.3±0.5 | 9.0±0.5 | 5.7±0.9 | 40.0±1.6 | 25.9±0.7 | 24.5±0.5 | 24.5±0.3 | 21.4±0.3 | 21.0±0.1 | 61.1±3.2 | 60.9±0.3 | 59.5±0.3 |
| ASER Shim et al. (2021) | 33.8±1.1 | 24.8±0.5 | 13.8±0.4 | 71.1±1.8 | 59.1±1.5 | 50.4±1.5 | 25.0±0.2 | 12.2±1.9 | 13.2±0.1 | 65.7±0.7 | 64.2±0.2 | 62.2±0.1 |
| DualNet Pham et al. (2021) | 9.8±0.3 | 5.0±0.3 | 3.8±0.2 | 38.5±0.4 | 32.1±0.5 | 25.2±0.4 | 20.1±0.3 | 12.2±1.9 | 7.5±0.2 | 20.7±0.5 | 16.2±0.3 | 14.7±0.3 |
| GDumb Prabhu et al. (2020) | 10.3±0.1 | 6.2±0.1 | 4.8±0.2 | 26.5±0.5 | 24.5±0.2 | 18.9±0.4 | 16.7±0.5 | 17.6±0.2 | 16.8±0.4 | 15.9±0.5 | 14.6±0.3 | 11.7±0.2 |
| SCR Mai et al. (2021b) | 10.7±0.1 | 4.7±0.1 | 4.0±0.2 | 41.3±0.1 | 31.5±0.2 | 24.7±0.4 | 17.5±0.2 | 11.6±0.5 | 5.6±0.4 | 19.4±0.3 | 15.4±0.3 | 14.9±0.7 |
| DER++ Buzzega et al. (2020) | 25.0±0.3 | 7.3±0.3 | 6.6±1.2 | 30.1±0.8 | 31.8±2.5 | 18.7±3.4 | 43.4±0.2 | 44.0±1.7 | 25.8±3.5 | 67.2±1.7 | 63.6±0.3 | 55.2±0.7 |
| IL2A Zhu et al. (2021a) | 8.7±0.1 | 7.2±0.1 | 4.1±0.1 | 36.0±0.2 | 32.1±0.4 | 29.1±0.4 | 24.6±0.6 | 12.5±0.7 | 20.0±0.5 | 65.5±0.7 | 60.1±0.5 | 57.6±1.1 |
| Co$^2$L Cha et al. (2021) | 14.7±0.2 | 7.1±0.1 | 3.1±0.1 | 32.0±0.1 | 21.0±0.3 | 16.9±0.2 | 16.9±0.4 | 16.6±0.6 | 9.9±0.7 | 60.5±0.5 | 52.5±0.9 | 42.5±0.8 |
| SSILAhn et al. (2021) | 11.3±0.1 | 2.7±0.1 | 2.8±0.1 | 36.0±0.7 | 29.6±0.4 | 13.5±0.4 | 40.1±0.5 | 33.9±1.2 | 21.7±0.8 | 44.4±0.7 | 36.6±0.7 | 29.0±0.7 |
| OCMGuo et al. (2022) | 4.7±0.1 | 1.8±0.1 | 1.3±0.1 | 23.0±0.2 | 14.0±0.7 | 12.0±1.1 | 12.2±0.3 | 8.5±0.3 | 4.5±0.3 | 23.5±1.9 | 21.0±0.3 | 18.6±0.5 |
| IFO | 11.5±0.5 | 4.6±0.2 | 4.1±0.1 | 17.0±0.3 | 14.4±0.2 | 11.0±0.5 | 15.9±0.3 | 16.0±0.2 | 7.1±0.3 | 22.8±0.6 | 20.0±0.4 | 15.5±0.6 |
| IFO+OCM | **4.2**±0.1 | **1.0**±0.2 | **1.1**±0.1 | **16.3**±0.3 | **8.1**±0.1 | **2.0**±0.7 | **11.9**±0.5 | **8.3**±0.4 | **4.3**±0.2 | **22.5**±0.3 | **18.6**±0.5 | **13.5**±0.8 |

Table 6: Test accuracy in the robustness benchmark. All numbers are the averages of 15 runs

| Dataset | CIFAR100-C | | | | | Tiny-ImageNet-C | | | | |
|---|---|---|---|---|---|---|---|---|---|---|
| Methods | Severity 1 | Severity 2 | Severity 3 | Severity 4 | Severity 5 | Severity 1 | Severity 2 | Severity 3 | Severity 4 | Severity 5 |
| OCM | 38.5±0.2 | 34.9±0.4 | 32.0±0.4 | 28.9±0.7 | 24.1±0.3 | 13.7±0.2 | 11.6±0.5 | 9.0±0.3 | 6.7±0.3 | 5.0±0.3 |
| IFO | 46.5±0.2 | 42.5±0.3 | 39.4±0.5 | 35.9±0.4 | 30.1±0.7 | 14.3±0.3 | 12.1±0.3 | 9.6±0.2 | 7.1±0.2 | 5.6±0.2 |

## C  ADDITIONAL RESULTS FOR THE DISJOINT ONLINE CL SETTING

We also adapted 4 batch/offline CL systems to online CL systems. Their results are given in Table 4. Our proposed IFO+OCM method still outperforms these batch CL baselines in the online CL setting.

**Forgetting rate.** From this forgetting rate table (Table 5), we observe an obvious drop in forgetting rate from OCM to IFO+OCM. In the ImageNet setting, the forgetting rates for the four top methods are 11.47 (SCR), 12.1 (SSIL), 11.7 (OCM), and 10.9 (IFO+OCM).

**Training time.** The training of our strongest method IFO+OCM is slower than that of OCM (the best baseline) (e.g., 4.5 percent on CIFAR100 dataset), but our accuracy performance is obviously better and it is critical to learn invariant features. We believe that improved training algorithms and advanced GPUs will relieve this issue in the future.

## D  EFFECTIVENESS OF INVARIANT REPRESENTATION LEARNING

Although better results of our method in the disjoint online CL setting have already indicated that our proposal is able to learn invariant features better. Here we use one additional experiment to further evaluate the effectiveness of learning invariant features: model robustness on unseen environments.

**Model robustness on unseen environments**. To verify that our method has learned invariant features to form the class representation, we conduct the following experiment (this is not a continual learning setting). After training on the CIFAR100 CIL setting, we test the trained model on the CIFAR100-C dataset Hendrycks & Dietterich (2019b), which is a model robustness benchmark consisting of 19 corruption types with five levels of severity applied to the original test set of CIFAR100. The corruptions come from four main categories: noise, blur, weather, and digital. Each

Table 7: Test accuracy in the unseen environment. All numbers are the averages of 15 runs. 'Art-painting' means that the model first learns the other three environments sequentially (order: *Cartoon* → *Photo* → *Sketch*), and then model is tested on the data points of the environment 'Art-painting'. 'Cartoon' means that the model first learns the other three environments sequentially (order: *Art-Painting* → *Photo* → *Sketch*), and then the model is tested on the data points of the environment 'Cartoon'. 'Photo' means that the model first learns the other three environments sequentially (order: *Art-Painting* → *Cartoon* → *Sketch*), and then the model is tested on the data points of the environment 'Photo'. 'Sketch' means that the model first learns the other three environments sequentially (order: *Art-Painting* → *Cartoon* → *Photo*), and then the model is tested on the data points of the environment 'Sketch'.

| Method | Art-painting | Cartoon | Photo | Sketch |
|--------|--------------|---------|-------|--------|
| $\mathcal{L}_{ce}$ | $12.7_{\pm 0.3}$ | $15.8_{\pm 0.5}$ | $12.5_{\pm 0.2}$ | $12.1_{\pm 0.3}$ |
| $\mathcal{L}_{inv}$ | $13.5_{\pm 0.2}$ | $17.4_{\pm 0.3}$ | $13.3_{\pm 0.4}$ | $13.2_{\pm 0.6}$ |
| $\mathcal{L}_{OCM} + \mathcal{L}_{ce}$ | $14.7_{\pm 0.5}$ | $17.8_{\pm 0.2}$ | $13.2_{\pm 0.6}$ | $13.1_{\pm 0.4}$ |
| $\mathcal{L}_{OCM} + \mathcal{L}_{inv}$ | $\mathbf{15.0_{\pm 0.4}}$ | $\mathbf{18.9_{\pm 0.2}}$ | $\mathbf{14.6_{\pm 0.1}}$ | $\mathbf{17.2_{\pm 0.4}}$ |

corruption has five levels of severity and "5" indicates the most corrupted one. Those corruptions are not used in training, so the test can be viewed as a test of the trained model in unseen environments. A model that has learned invariant features should achieve higher performance. From Table 6, we observe that IFO indeed outperforms the OCM method on 19 unseen environments. Note that we use IFO as we want to isolate effectiveness of the invariant feature learning, which is more important for unseen environments. The gap between IFO and OCM gets larger as the level of severity increases. We conduct a similar experiment on the Tiny-ImageNet-C dataset (another robustness dataset in Hendrycks & Dietterich (2019b)), and the conclusion is consistent (Table 6). Those experiments empirically verify that our method learns more invariant features than OCM.

## E    MORE DETAILS ABOUT THE I-BLURRY ONLINE CL SCENARIO

Following Koh et al. (2021), for all methods, we use the batch size of 16 and 3 updates per streamed sample for CIFAR100 and the batch size of 32 and 3 updates per streamed sample for TinyImageNet, and employ ResNet-34 for CIFAR100 and TinyImageNet. In the CIFAR100 setting, each task has 10 unique disjoint classes and 10 dominant blurry classes exclusively. The selection process of the classes for each task is random. In the TinyImageNet setting, each task has 20 unique disjoint classes and 20 dominant blurry classes exclusively. AutoAugment Cubuk et al. (2019) and CutMix Yun et al. (2019) are also used as data augmentations. For CLIB and CLIB+INV, we use the same adaptive learning rate schedule Koh et al. (2021) with $\gamma = 0.95$ and $m = 10$ for the two datasets. We use their official code `https://github.com/naver-ai/i-Blurry` to run the experiments.

## F    MORE DETAILS ABOUT THE DATA ENVIRONMENT SHIFT ONLINE CL SCENARIO

Here we cannot use MNIST, CIFAR, and ImageNet datasets as they do not have clearly defined environments. We use the PACS dataset Li et al. (2017) to simulate this scenario. PACS has four different environments: art painting, cartoon, photo, and sketch. Each environment has data points of the same seven classes. We choose each set of three environments as the training environment and the remaining environment as the test environment. Thus four experiments are conducted with different training and test environments. We report the average test result over the four experiments as the empirical estimation of the ability of our method in learning invariant features. We use ResNet-18 (not pre-trained) as the backbone for our method IFO and baselines and use the Adam optimizer and set the learning rate as 1e-3 for all methods. The batch size for the new data batch is 10 and there is no need to store or sample buffer data. Data of each environment is run in one epoch for online CL. The core of this setting is to learn invariant features under the shift of environments. So we focus on the representation-learning loss and do not consider/use the replay strategy to overcome forgetting. From Table 7, we observe that the model optimized with our invariance loss achieves higher test performance than that of the traditional cross-entropy loss in the unseen environment.

Table 8: Ablation accuracy for the hyper-parameters - an average of 5 runs. $B$ is the memory buffer size.

| Dataset | $s$ | | | | | | | $r_1$ | | | | | | | |
|---|---|---|---|---|---|---|---|---|---|---|---|---|---|---|---|
| | 0 | 1 | 2 | 3 | 4 | 5 | 6 | 0.125 | 0.25 | 0.375 | 0.5 | 0.625 | 0.75 | 0.875 | 1 |
| CIFAR100 ($B$=2k) | 35.1±0.7 | 36.6±0.3 | 37.1±0.3 | 37.6±0.4 | 38.0±0.2 | 38.1±0.5 | 38.1±0.2 | 30.2±0.2 | 34.1±0.1 | 35.2±0.4 | 37.3±0.4 | 38.0±0.3 | 38.1±0.5 | 37.8±0.2 | 37.0±0.2 |
| TinyImageNet ($B$=2k) | 12.0±0.2 | 12.3±0.2 | 12.5±0.2 | 13.1±0.3 | 13.0±0.4 | 13.5±0.5 | 13.7±0.3 | 9.0±0.5 | 11.5±0.3 | 11.7±0.5 | 12.9±0.3 | 13.3±0.2 | 13.5±0.5 | 12.9±0.5 | 12.6±0.3 |

Table 9: Ablation accuracy for the hyper-parameters - an average of 5 runs. $B$ is the memory buffer size.

| Dataset | $k$ | | | | |
|---|---|---|---|---|---|
| | 1 | 2 | 3 | 4 | 5 |
| CIFAR100 ($B$=2k) | 37.8±0.1 | 37.9±0.2 | 38.1±0.5 | 38.1±0.2 | 38.0±0.3 |
| TinyImageNet ($B$=2k) | 12.4±0.3 | 12.8±0.3 | 13.5±0.5 | 13.0±0.4 | 13.1±0.3 |

## G ABLATION ANALYSIS OF HYPER-PARAMETERS IN IFO

Based on the results in Table 8. we set $s$ in $Aug_{color}$ to 5 as it achieves the best performance with less computation. Also, we set $r_1$ in $Aug_{add}$ as 0.75 and the number of clusters $k$ as 3 (Table 9).

## H INFLUENCE OF LEARNING INVARIANT FEATURES

To further investigate how learning invariant features helps the performance of CL methods, we measure three abilities of OCM and IFO+OCM: **(1)** the ability to establish decision boundaries between the classes within the new task by recording the accuracy performance of the new task. From Figure 3(a), we see that IFO+OCM's accuracy of the new task outperforms that of OCM as learning invariant features makes the model improve its generalization power. **(2)** the ability to maintain the learned decision boundaries within a task by calculating the average task incremental accuracy of the previous tasks. Our method IFO+OCM again outperforms OCM (Figure 3(b)) as IFO+OCM mitigates the overfitting problem of the limited buffer data. **(3)** the ability to establish class boundaries across tasks. We measure this by considering only the logit of the true label of each test instance in a task and the logits of the classes from the other tasks when the model predicts the label of the test instance. IFO+OCM's performance is still better than that of OCM (Figure 3(c)) as IFO+OCM reduces variant features, which enables the model to project those class representations to different locations in the space, making the decision boundaries easier to establish. Also from Figure 3 also shows that improving the last ability is the biggest challenge for online CL.

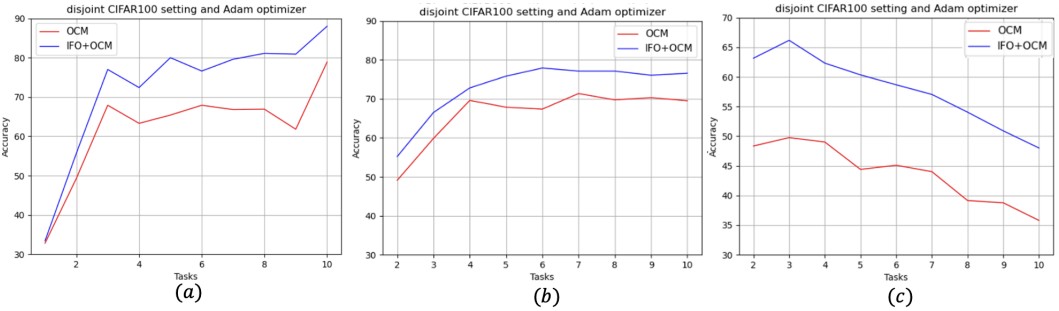

Figure 3: We conduct experiments on the CIFAR100 dataset (10 tasks) with a buffer size of 2k. (a) Performance of learning new tasks. (b) the performance of maintaining learned decision boundaries. (c) Performance of establishing decision boundaries across different tasks.

## I  LIMITATIONS AND BROAD IMPACT

This paper has shown the necessity of learning invariant features for overcoming forgetting and has demonstrated strong performances across multiple datasets and settings. Given that much of the research in continual learning (CL) has focused on designing empirical algorithms, the formal study in this paper should have a broader and deeper impact. However, we have not yet identified sufficient conditions for feature learning in order to eliminate forgetting in a CL system. Finding the set of principles that can guide feature learning in CL is of utmost importance. Our future work will be dedicated to investigate this problem. Another limitation is that our paper focuses on online CL settings. In our future work, we will also try to adapt our method to the batch CL setting.

