# OpenReview forum: "Invariance as A Necessary Condition for Online Continual Learning"
_ICLR.cc/2024/Conference — ICLR 2024 Conference Withdrawn Submission_

### Official Review · Reviewer_yhUR · 2023-10-25

**Soundness:** 2 fair
**Presentation:** 1 poor
**Contribution:** 1 poor
**Rating:** 3
**Confidence:** 4

**Summary:**

This paper addresses the challenge of online continual learning, emphasizing the necessity of learner invariance to prevent catastrophic forgetting. The authors specifically focus on achieving invariance with respect to background and edge expansion, considering them as crucial target variables.

The proposed solution, IFO, is introduced as an approach to address this issue. However, in direct comparison with the competing baseline, OCM, IFO underperforms. Despite this, it is important to note that IFO complements OCM from an alternative perspective, providing a contribution to the field of continual learning.

**Strengths:**

S1:
This paper delves into a relatively unexplored realm within continual learning: the concept of invariance.

S2:
The authors offer a theoretical foundation for their argument, grounded in the principles of causality. However, I am not 100% sure about the validity of their findings, especially regarding the gradient norm.

S3:
The authors conduct evaluations across various distinct class-incremental learning scenarios, including settings involving blurry and disjoint data, enhancing the comprehensiveness of their study.

**Weaknesses:**

W1:
The experimental contribution of this paper appears to be limited. Despite the central argument emphasizing the necessity of invariance for online continual learning, the experimental evidence provided is weak. Multiple instances indicate that the proposed approach, IFO, often underperforms the compared baseline, OCM. This suggests that a holistic approach might be more critical than mere invariance in preventing forgetting.

W2:
Regarding the technical contribution, the proposed IFO relies on an alignment loss, essentially a variant of the contrastive learning loss. This contribution, while present, seems limited in its originality and novelty.

W3:
The scope of the studied invariance setting appears to be constrained. The choice to generate foreground-background blendings, where the specific nuisance to be addressed is known, might be considered somewhat artificial. This artificiality raises questions about the broader applicability and relevance of the findings.

W4:
In terms of presentation, the quality of writing is notably low. The paper proves challenging to read due to unsupported claims, repetitiveness, excessively long sentences, and subpar figures and table presentation. These issues significantly impact the clarity and overall readability of the paper.

Considering these concerns, the paper is currently not deemed suitable for publication until these issues are addressed and improved in the rebuttal phase.

**Questions:**

N/A

---

> ### Author Response · Authors · 2023-11-15
> **A brief response to the weaknesses**
>
> W1:The experimental contribution of this paper appears to be limited. Despite the central argument emphasizing the necessity of invariance for online continual learning, the experimental evidence provided is weak. Multiple instances indicate that the proposed approach, IFO, often underperforms the compared baseline, OCM. This suggests that a holistic approach might be more critical than mere invariance in preventing forgetting.
>
> We believe that this is a misunderstanding. Our paper does not claim that learning invariant features is sufficient for online CIL and learning holistic representation is not necessary. In fact, our study emphasizes that learning invariant features is necessay and it indeed improves the replay-based method's performance by a clear margain. Also, learning invariant representation (our work) is complementary to the learning of holistic representation (OCM) as the model needs to learn class invariant features as many as possible in online CIL. Our empricial result also showed that IFO+OCM outperforms any one of the two methods.
> Empirical results are important, but we would also like to emphesize that as a coommunity, we need to have a theoretical understanding of the types of features that should be learned in CIL. OCM showed that the holisticness is necessary. We showed that invarance is also necessary. No other previous work has study these necessary conditions.
>
> W2: Regarding the technical contribution, the proposed IFO relies on an alignment loss, essentially a variant of the contrastive learning loss. This contribution, while present, seems limited in its originality and novelty.
>
> Our method is not a variant of the contrastive learning loss. Our loss is an approximation of the Interventional Empirical Risk in Section 5.4. And the contrastive loss is an approximation of the mutual information between positive samples. Also, our environmental augmentation changes the original environment of the image while maintaining its semantic meaning. However, the traditional augmentation for constructing the positive samples in contrastive loss (e.g., random resized crop) can destroy image's original meaning.
>
> Again, we would like to stress the theoretical contribution of identifying feature invariance as necessay for CIL, which is important because it gives a theoretical guidance in the design of empirical algorthms. So far, little theoretical work has been done in terms of feature learning for CIL.
>
> W3 The scope of the studied invariance setting appears to be constrained. The choice to generate foreground-background blendings, where the specific nuisance to be addressed is known, might be considered somewhat artificial. This artificiality raises questions about the broader applicability and relevance of the findings.
>
> It's notoriously hard to automatically identify all variant features in natural images. Thus, we propose two simple environmental augmentations based on prior knowledge as the first try. We also verify our method's effectiveness on three different online continual learning scenarios and 6 datasets.
>
> W4 In terms of presentation, the quality of writing is notably low.
>
> We are sorry for this. We have polished the whole paper to improve the quality of the writing.

---

### Official Review · Reviewer_C77j · 2023-10-29

**Soundness:** 3 good
**Presentation:** 3 good
**Contribution:** 3 good
**Rating:** 6
**Confidence:** 4

**Summary:**

This paper highlights the importance of learning invariant features in mitigating the phenomenon of catastrophic forgetting in class-incremental continual learning through theoretical analysis.  A new method based on experience replay is then proposed for learning invariant features via creating environmental variants using 3 data augmentation techniques.

**Strengths:**

1. This paper underscores the importance of learning invariant features in mitigating the phenomenon of catastrophic forgetting in class-incremental learning, thus providing a promising direction for the development of novel CIL algorithms.

2. Despite the existence of prior research on the idea of learning invariant features in continual learning, this paper offers a review on the related works and places itself in a good position in the literature.

3. The integration of the proposed method with OCM results in exceptional empirical performance.

**Weaknesses:**

1. The proposed methods incorporates many hyper-parameter, including $\alpha$, $s$, $k$, $r_1$ in data augmentation. However, determining the optimal values for these hyper-parameters in continual learning can be challenging. Furthermore, this paper lacks comprehensive studies of the effects of the hyper-parameters and their determination.

2. The paper does not empirically verify the individual impact of learning invariant features in mitigating catastrophic forgetting. It is always combined with replay. Moreover, the accuracy and forgetting of the proposed IFO are not competitive compared to OCM and its optimal performance relies on integration with OCM.

3. Some minor issues:
- No experimental results on time efficiency comparison in the Appendix.
- $\theta$ is repeatedly defined as feature extractor parameter and as binary vector in Eqn. (2).

**Questions:**

1. How is the integration of IFO and OCM achieved? Does it involve the direct addition of two losses during model training? Furthermore, how does IFO coalesce with OCM in feature learning? While the objective of OCM is to learn as many features as possible, IFO solely focuses on invariant features, leading to some conflicting goals.
2. For background color augmentation, during the initial stages of training, when the classifier is not yet fully trained, there may be concerns regarding the accuracy of background color augmentation. There may be concerns where CAM fails to correctly identify the background, resulting in incorrect augmentation.
3. For method II proposed in 5.3, as mentioned, "We collect all new task data already stored in the buffer". Method II seems not applicable if there is no new task data in the buffer, which is often the case in continual learning when new task appears. Is this correct? Also, when the buffer is updated, do the k clusters get updated?

---

> ### Author Response · Authors · 2023-11-15
> **A brief response to your questions**
>
> Question1. How is the integration of IFO and OCM achieved? Does it involve the direct addition of two losses during model training? Furthermore, how does IFO coalesce with OCM in feature learning? While the objective of OCM is to learn as many features as possible, IFO solely focuses on invariant features, leading to some conflicting goals.
>
> Yes, we directly optimize the combination of the two losses in model training. OCM tries to maximize the mutual information between the image and the hidden representation by learning as many features as possible. However, OCM may introduce variant features. IFO discourages the learning of variant features but it does not control how many invariant features should be learned. With the combination of IFO and OCM, the model learns class invariant features as many as possible. We believe this should be the correct direction for representation learning in online CIL.
>
> Question2: For background color augmentation, during the initial stages of training, when the classifier is not yet fully trained, there may be concerns regarding the accuracy of background color augmentation. There may be concerns where CAM fails to correctly identify the background, resulting in incorrect augmentation.
>
> During the initial stages of training, CAM may fail to correctly identify the object. But our augmentation would not distort other core invariant class features (e.g., the shape feature). Thus our method potentially makes model learn these non-color invariant features first and then recover the learning of class color in the later training stage.
>
> Question 3. For method II proposed in 5.3, as mentioned, "We collect all new task data already stored in the buffer". Method II seems not applicable if there is no new task data in the buffer, which is often the case in continual learning when new task appears. Is this correct? Also, when the buffer is updated, do the k clusters get updated?
>
> In online CIL, the environment in the initial training stage is very different from that in the later stage. We can use the early environment to augment the newest training batch. But the model can only visit each batch one time. So we design the method II to utilize the early stored buffer data of the same task to augment the training data. If there is no new task data in the buffer, that means the model is still in the initial stage of training. And we do not need method II to augment the training data. When the buffer is updated, we update the k clusters.

---

### Official Review · Reviewer_Xe1e · 2023-10-30

**Soundness:** 2 fair
**Presentation:** 3 good
**Contribution:** 2 fair
**Rating:** 5
**Confidence:** 4

**Summary:**

Previous studies in this direction have shown that learning holistic representations in Continual Learning settings can help to mitigate forgetting. However, in this paper, the authors argue that the representations must also be invariant. This paper shows, theoretically and empirically, that having both holistic and invariant representations helps in online scenarios. After raising the need to remove spurious correlation of the previously learned model, the authors propose a new method for Online Continual Learning. They show good performance in traditional disjoint task scenarios in multiple benchmarks and also in blurry task boundaries and data shifts.

**Strengths:**

- Studying the theory behind a scenario and understanding the how and why of the problems is an excellent way of proposing new methods.
    - It is more promising when results on the empirical side also accompany these findings.
- Evaluating the method in various online scenarios proves its robustness.
- The augmentations proposed are interesting. Increasing the variability of the data can help improve the generalization capabilities of the model.
    - These augmentations can work better than MixUp, under specific settings.

**Weaknesses:**

- My main concern is the Theoretical Analysis, which can have significant repercussions in the practical section. I agree with the intuition of the authors that if we focus on learning invariant and holistic representations, we can mitigate forgetting. However:
    - To learn these representations, the authors assume that the model converges. Something that is not the case in most Online scenarios. The paper does not show that the model learns invariant or spurious correlations. Showing that a model learns these kinds of representations is a challenging task since there is a whole research area that is focusing in this direction.
    - Assuming that one can select which representations are invariant or spurious is a big If, more with the limited distributions that are stored in the buffer. One may obtain an approximation, but solving this problem is not trivial, as multiple work in spurious correlation has proven.
    - I appreciate the great work done on the theoretical analysis of the work, together with the motivations and intuitions. However, the demonstrations and assumptions are not without issues.
- There are a few mentioned in the paper that the proposed method works in Class-Incremental Learning settings, without specifying the online setting. However, it could be better to refer to Online Learning. Changing between class-incremental and online class-incremental and use both interchangeable could create confusion.
- The augmentation methods proposed only work in particular scenarios. Where is one easy-to-detect object and is centered in the image. Something that occur in limited opportunities.
    - Also, as mentioned in the paper, the authors assume that the background is the primary source of spurious correlation. This limits the proposal even more.
- The paragraph “Learning invariant representations” in the results section is incomplete. It mentioned that the model was trained with original datasets, and I would assume until convergence. Training the model this way could create completely different representations. It is essential to show that the proposed method in Online Learning can generate those representations, something that, as mentioned, I am not entirely sure about.

**Questions:**

- Please explain the last paragraph of Appendix A. There are a few steps that are not trivial to me.
- Can we have a CL method that works under your assumptions but without a memory buffer? Or is access to a memory essential to achieve a model that does not forget?
- What are the similarities or differences between the “data environment shift setting” and a “domain incremental”?
- One significant limitation of Online scenarios is the computational capacity. Some argued that most methods underperform due to the low time to train the model. However, in the proposal, you are increasing the losses, by increasing the augmentations (Eq 1, 5 and 7). How much do you expand the batch size to be able to see all the samples? Or do you keep the batch size fixed and increase the interactions you train the model?
    - How much does the computational time increase?
- The size of the buffer and the samples stored can significantly influence the accuracy obtained in memory-based methods. How does the K value behave when changing the buffer size?
- Can you explain Table 3 in a different way? When it said, “no align”, you just removed the align? In most cases, the difference is less than 1% of accuracy. How can you identify if one of these components is not necessary?

---

> ### Author Response · Authors · 2023-11-15
> **A brief response to your questions**
>
> Question 1: Please explain the last paragraph of Appendix A. There are a few steps that are not trivial to me.
> We update the paper and add detailed steps in the last paragraph of Appendix A. Please check it.
>
> Question 2: Can we have a CL method that works under your assumptions but without a memory buffer? Or is access to a memory essential to achieve a model that does not forget?
>
> Our work aims to address the theoretical question of what types of features or representations are necessary and should be learned in CIL. Our method (IFO) requires a memory buffer to safeguard the features learned from previous classes. Our method is an online CL method and it is hard to do online CL without a memory buffer. We are unaware of any existing online CL method that does not use a memory buffer. We have revised the introduction to reduce confusion.
>
> Question 3: What are the similarities or differences between the “data environment shift setting” and a “domain incremental”?
>
> In general, data environment shift can be seen as a form of domain incremental learning with the classes across different tasks remaining the same, but the environments differ. However, traditional vision domain incremental setting like Rotate-MINISTsimply rotate the digit by a random degree. Our data environment shift setting is more challenging because it shifts the entire data environment for each task (Art-painting -> Cartoon -> Photo -> Sketch).
>
> Question 4: One significant limitation of Online scenarios is the computational capacity. Some argued that most methods underperform due to the low time to train the model. However, in the proposal, you are increasing the losses, by increasing the augmentations (Eq 1, 5 and 7). How much do you expand the batch size to be able to see all the samples? Or do you keep the batch size fixed and increase the interactions you train the model?How much does the computational time increase?
>
> With data augmentation, the training batch size increases by seven-fold. We train the model with all augmented data points in one iteration. This increases the training time from 25 minutes (using the experience replay method) to 1 hour and 12 minutes (with our IFO method) on the CIFAR-100 dataset. We usean A100 GPU in training. However, our method outperforms the experience replay method  in accuracy by a significant margin. We anticipate that more advanced GPUs in the future will reduce the training time.
>
> Question 5. The size of the buffer and the samples stored can significantly influence the accuracy obtained in memory-based methods. How does the K value behave when changing the buffer size?
>
> We've observed that the results with different choices of k are consistent across various buffer sizes. For instance, when the buffer size is set to 1k on the CIFAR100 dataset, the performances for k values from 1 to 5 are 33.2, 33.2, 34.0, 34.5, and 34.7, respectively. Similarly, with a buffer size of 5k on the CIFAR100 dataset, the corresponding performances for k from 1 to 5 are 43.2, 45.5, 46.5, 46.5, and 46.6 respectively. As a result, we decided to set the value of k to 3.
> In the scenarios where the buffer size is very small (e.g., 10), the buffer may not store a large amount of previous training data for the same task in online class incremental learning (CIL). In such cases, we omit the use of the second method for environmental augmentation. Consequently, there's no need for the selection of k.
>
> Question 6. Can you explain Table 3 in a different way? When it said, “no align”, you just removed the align? In most cases, the difference is less than 1% of accuracy. How can you identify if one of these components is not necessary?
>
> For the "no align" experiment, we continue to employ the environmental augmentation to enhance the training data; however, we refrain from aligning the representations of the same image with different environmental augmentations. Consequently, the model may learn features related to the environment rather than an invariant class representation for classification. We identify unnecessary components by assessing whether they result in a drop in validation performance or fail to yield a noticeable improvement in validation performance across different datasets.

---

### Author Response · Authors · 2023-11-15
**A reminder about the revised paper.**

Hi reviewers and AC

We have revised and updated our paper. All edited places are colored with blue.